

# Human–robot interaction: predicting research agenda by long short-term memory

Jon Borregan-Alvarado[1], Izaskun Alvarez-Meaza[1], Ernesto Cilleruelo-Carrasco[1] and Rosa Maria Rio-Belver[2]

[1] Industrial Organization and Management Engineering Dept., University of the Basque Country UPV/EHU, Bilbao, Biscay, Spain
[2] Industrial Organization and Management Engineering Dept., University of the Basque Country UPV/EHU, Vitoria, Araba, Spain

## ABSTRACT

The article addresses the identification and prediction of research topics in human–robot interaction (HRI), fundamental in Industry 4.0 (I4.0) and future Industry 5.0 (I5.0). In the absence of research agendas in the scientific literature, the study proposes a multilayered model to create a precise agenda to guide the scientific community in new developments in collaborative robotics and HRI technologies. The methodology is divided into four stages, which make up the three layers of the model. In the first two stages, scientific articles on HRI for the period 2020–2021 were collected and analyzed using data mining techniques together with VantagePoint and Gephi software to identify keywords and their relationships. These initial stages form layer 1 of the model, where the main scientific themes are recognized. In the third stage, article titles and abstracts are cleaned and processed using natural language processing (NLP) techniques, generating word embeddings models that highlight relevant HRI-related terms, forming layer 2. The fourth and final stage uses Recurrent Neural Networks (RNN) with long short-term memory (LSTM) architecture to predict future topics, consolidating the previously identified terms and forming layer 3 of the model. The results show that in layer 1 HRI has intensive application in various sectors through advanced computational algorithms, with trust as a key feature. In layer 2, terms such as vision, sensors, communication, collaboration and anthropomorphic aspects are fundamental, while layer 3 anticipates future topics such as design, performance, method and controllers, essential to improve robot interaction. The study concludes that the methodology is effective in defining a robust and relevant research agenda. By identifying future trends and needs, this work fills a gap in the scientific literature, providing a valuable tool for the research community in the field of HRI.

Corresponding author
Jon Borregan-Alvarado,
jon.borregan@ehu.eus

## INTRODUCTION

In order to provide a clear and concise introduction, it has been divided into several sub-sections. In each of these, an introductory theme is presented sequentially.

## I4.0/I5.0 & collaborative robots

Over the last 300 years, industry has undergone several industrial and technological revolutions. The first industrial revolution began at the beginning of the 18th century, where water and steam power drove mechanized systems. Then, at the end of the 19th century, the second industrial revolution emerged, where, thanks to electricity, companies started with mass production, most notably in the automotive sector. This second industrial revolution was the predecessor of the third industrial revolution in the 1970s, where the rise of computing and robotics developed automated processes (*Colombo et al., 2021*). In 2011, the fourth industrial revolution was born, where humans and things are connected in both the physical and virtual world through the internet of things (IoT); cloud computing (CC); artificial intelligence (AI); cyber physical systems (CPS); robotics; augmented/virtual reality (AR&VR) (among others *Chung, 2021*).

The different industrial revolutions have been accompanied by different scientific and technological plans called society 1.0, society 2.0, *etc.* (*Elim & Zhai, 2020*), Society 5.0 having already been implemented in 2016 by Japan, the objective of which is the improvement of productivity, competitiveness, connection and welfare; maximizing human use in such technological transformation (*Narvaez Rojas et al., 2021*). Moreover, this Society 5.0 is in line with the main positioning of industry 5.0, which focuses on the human being associated with issues of sustainability, human–robot interaction (HRI) and resilience (*Madsen & Berg, 2021*).

This last concept, HRI, together with different forms of collaboration, is now leading the way in different industrial applications where collaborative robots (cobots) stand out as one of the main elements (*Schmitt et al., 2021*). Therefore, it is not surprising, that the term industry 5.0 refers to human–robot-intelligent machine labor collaboration, creating a human-centric environment where robots interact and help work faster and better, eliminating painful and non-value-adding tasks for humans, among others (*Martin et al., 2021*). Moreover, being highly reliable and programmable, it is possible to meet market demands and customer needs in a flexible and customized way, increasing quality and efficiency, while improving production line flow and reducing final costs and risks (*Javaid & Haleem, 2020*).

Collaborative robots also bring a human touch to industry 5.0: robot design and features nowadays tend to replicate human-like mental models (anthropomorphization), such as descriptive linguistics, movement style or appearance, in order to gain increased trust during human–robot work interaction (*Kopp, Baumgartner & Kinkel, 2022*).

Despite the advantages provided, it should be stressed that the main function of using collaborative robots is to enhance or complement human capabilities and work, not replace them, since, as mentioned above, one of the pillars of industry 5.0 is the generation of a human-centered environment (*Coronado et al., 2022*).

As detailed so far, the human-centered manufacturing paradigm of industry 5.0 involves collaborative robots interacting with humans to perform complex and/or dynamic tasks (*Li et al., 2023*). Accordingly, HRI is booming thanks to Industry 4.0 and the future industry 5.0, being one of the pillars of this latest industrial revolution.

## HRI concept

The HRI concept refers to a field which aims to improve the interaction between humans and robots in different activities (*Azmin, Shamsuddin & Yussof, 2016*), and these activities can be performed in robotic environments such as industrial, space, agriculture and forestry, construction, hazardous applications, mining, disaster, surveillance and security, medical and surgery, automotive, rehabilitation and health care, domestic, competitions and challenges, social, and educational, among others (*Siciliano & Khatib, 2016*). The improvement of this interaction is subject to prior design work, which, in turn, is conditioned and influenced by different factors such as social, organizational, physical, cognitive, economic and environmental, among others (*Simões et al., 2022*). Furthermore, human–robot interaction is composed of different categories depending on the degree of collaboration/interaction. For example, the author *Zacharaki et al. (2020)* include HRI in four categories, namely:

- Coexistence: human and robot work in concert without sharing a work area
- Synchronized: human and robot work alternately within the same work area
- Cooperation: human and robot do not work simultaneously on the same element, although they share the same work area
- Collaboration: human and robot work simultaneously on the same element and in the same work area.

In contrast, other authors such as *Hentout et al. (2019)* encompass it in three main categories, with minor variations in their description:

- Coexistence: human and robot share a work environment sequentially, and therefore without the need for mutual contact, thus avoiding collisions.
- Cooperation: human and robot work together towards the same goal, sharing the workspace simultaneously. Collaboration involves identifying and avoiding possible collisions.
- Collaboration: human and robot perform a complex task with direct interaction, which can be subcategorized into non-contact collaborations and physical collaborations, the latter concept giving rise to the term physical human–robot interaction (pHRI).

Therefore, pHRI, a noteworthy category belonging to HRI, should be designed to adapt to human movements, avoiding collisions and ensuring both safety and efficiency (*Mukherjee et al., 2022*), limiting the speed of movement in human–robot collaboration (HRC) (*Qi, Song & Dai, 2022*).

## Motivation, contribution and benefits

The following subsection presents the state of the art corresponding to the field of HRI in terms of research agendas. In addition, the motivation, contribution and benefits derived from this scientific work are justified by what is described and observed through the results obtained.

### Lack of HRI research agendas

Regarding research works in scientific literature, it has been observed that there are no research papers defining research agendas related to HRI. In order to justify the above conclusion, we designed and applied different queries, and thus observed the existence or non-existence of scientific works similar to our study.

### Search methodology

The model or basic query developed is based on the search for various concepts defined within the scientific works and indexed as author keywords. The use of author keywords in the search (AUTHKEY) ensures greater efficiency and better results in the return or retrieval of the information obtained through the query (*Lu & Kipp, 2014*), because the keywords have been selected by the authors themselves and not by the scientific database (as opposed to index keywords). The model or base query used was the following:

(AUTHKEY("human robot interaction\*") or AUTHKEY("human–robot interaction\*") or AUTHKEY("human–robot interaction\*")) AND (AUTHKEY("predict\*") or AUTHKEY("forecast\*") or AUTHKEY("future\*")) AND (TITLE-ABS-KEY("XXXXX"))

As can be seen, the last term of the query acts as an auxiliary term, and in order to cover a wider search field, the search by means of the auxiliary term has been extended to titles, abstracts and index keywords (TITLE-ABS-KEY), and not exclusively to author keywords.

### Auxiliary terms used

The nine auxiliary terms used were as follows:

- Python: A high-level programming language, and one of the most popular and widespread (*Narayanan, 2019*; *Javed et al., 2019*).
- RNN: Recurrent neural network is a type of neural network (NN) architecture primarily used to identify patterns in a dataset (*Schmidt, 2019*).
- LSTM: Long short-term memory is a type of RNN used in machine learning (ML) to process and predict by handling long sequences of data. (*Sherstinsky, 2020*; *Yoon et al., 2021*).
- NLP: Natural language processing is a discipline that combines artificial intelligence (AI) and Linguistics, and focuses on studying the problems of linguistic communication between humans and computers (*Jiang & Lu, 2020*; *Khurana, 2023*).
- tensor\*: Referring to TensorFlow or TensorBoard tools, a web tool used to interactively visualize the word embeddings of our model and render them in two or three dimensions (*Pandey, 2021*) for later mapping.
- Gephi\*: Referring to open source Gephi software, software used to construct graph-based maps (*Bastian, Heymann & Jacomy, 2009*).
- vantage\*: Referring to VantagePoint (VP) software, text mining software (*Macías-Quiroga et al., 2021*), which makes it possible to perform complex statistical analyses of the data (*Alvarez-Meaza et al., 2024*).
- vectorial: This refers to the vectorial distance (quantitative) of the term studied with respect to the rest, in order to identify those terms with greater similarity or affinity to the HRI concept.

- Distance: This refers to the vectorial distance (quantitative) of the term studied with respect to the rest, in order to identify those terms with greater similarity or affinity to the HRI concept.

  *: wildcard character with which to represent and cover any other term within our area of study, through the different existing possibilities of the character * onwards.

### HRI field, NLP techniques and LSTM networks

The following sub-section presents a state of the art regarding the contributions, advantages or benefits provided by NLP techniques and LSTM networks in comparison with other methods, thus justifying their choice within the methodological development. Furthermore, it also justifies the importance of this scientific study within the field of HRI, in order to help complement and provide answers to other existing scientific articles and more recent ideas within this field.

- HRI

  Part of the aim of this scientific work is to predict potential terms in the field of HRI, which may provide answers to the questions raised and corroborate the results of other scientific papers within this field. For example, *Zamboni & Valente (2020)*, highlights the need for advanced and proactive sensing technologies to enhance safe human–robot collaboration. Prediction of relevant terms can identify and anticipate technological developments that address these needs. *Tong, Liu & Zhang (2024)*, on the other hand, stress the importance of improving knowledge about biological motion, structural design and energy efficiency in humanoid robots. Our prediction model can pinpoint emerging terms in these fields, facilitating advances and continuous improvements. Finally, *Gold et al. (2007)*, mention the lack of common platforms and longitudinal studies. By predicting future terms and trends, this work can help establish a stronger and more uniform foundation for HRI research, fostering greater coherence and collaboration in the scientific community.

- NLP

  The natural language processing (NLP) technique is suitable for obtaining relevant terminology and predicting future terms in a specific field due to its ability to analyze large volumes of text and extract meaningful patterns. *Adnane, Fadoua & Salmane (2023)* highlight how NLP technologies can generate texts automatically and the importance of robust detection systems to preserve discourse integrity. This demonstrates the ability of NLP to analyze and manage large amounts of textually complex information, identifying relevant terms. On the other hand, *Joshi, Shinde & Das (2023)* show how NLP models improve performance through data augmentation techniques, indicating that NLP can adapt and continuously improve, allowing the prediction of future terms based on existing patterns. These capabilities make NLP a suitable tool compared to other techniques, due to its adaptability and accuracy in text processing and analysis.

- LSTM

  The recurrent neural network (RNN) with long short-term memory (LSTM) architecture is suitable for making future predictions due to its ability to learn and remember

long-term patterns in sequential data, outperforming other techniques in accuracy and efficiency. _Xu, Song & Hao (2022)_ show that, although shallow machine learning models can be fast, LSTM models offer superior accuracy, especially on unbalanced data. _Feng et al. (2022)_ indicate that, although process-based differentiable models approach the performance of LSTMs, LSTMs are still difficult to beat in terms of accuracy for intensively observed variables. Finally, _Choudhary (2023)_ compare classical time series prediction methods with deep learning models, concluding that LSTM models are more versatile and accurate in predicting sequential data, adapting better to different data sets. These characteristics make LSTM an ideal choice for future predictions in the field of HRI.

### Search results

Once the query was executed with each of the auxiliary terms, it was observed that the term "distance" yielded seven scientific papers corresponding to mathematical calculations of human–robot approximation, which we discarded because our term "distance" refers to vector distance calculations, and therefore to the study of similarity or affinity between terms in the HRI field. The auxiliary terms "Python", "NLP", "tensor", "Gephi", "vantage" and "vectorial" returned no results.

### Relevant works identified

The following three research papers focusing on the improvement of HRI processes using advanced algorithms such as neural networks have been identified under the auxiliary terms "RNN" and "LSTM":

- _Türker et al. (2018)_ propose an audiovisual prediction system for head nodding and turning events _via_ LSTM-RNN to create a more natural HRI system.
- _Wu, Zhong & Yang (2022)_ elaborate on a vision-based gesture prediction system with which to predict dynamic hand gestures through LSTM-RNN, and contribute to a more natural interaction.
- _Chellali & Li Chao (2018)_, in contrast, use LSTM, to derive the final position and timing of the human hand when performing a high-five game with robots, the interactions being key issues between robots and humans.

### Conclusion of the state of the art

Therefore, as indicated above, the literature review shows that there are no research papers defining research agendas related to HRI.

Consequently, and given the lack of research agendas, the main objective of the following scientific work is to identify and predict the research topics related to HRI in order to generate a research agenda, taking the scientific literature of the area as a source, and generating a multilayered model of analysis. In this way, the research agenda will allow research groups to define their strategies in a more accurate and focused way, which is a crucial advantage to maximize the efficiency and impact of their projects and research.

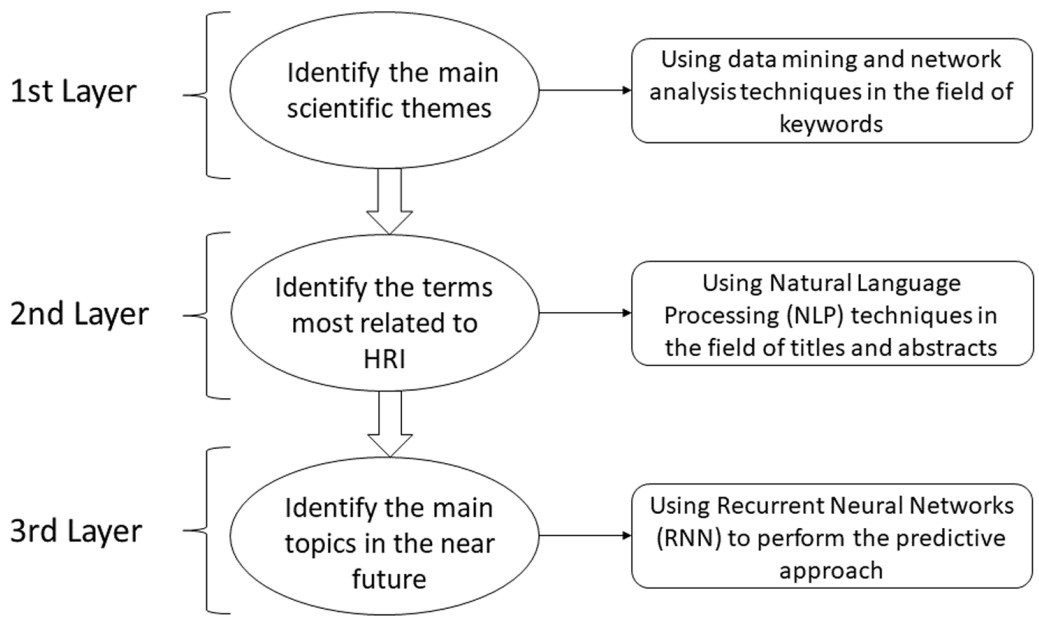

**Figure 1   Scheme referring to the different layers defined and used.**

## Multilayer model: research agenda

The different layers of the multilayer model mentioned above have been analyzed from different approaches through the combination and use of different methodologies. Then, after observing and analyzing the different results obtained by each of the layers, a joint analysis is performed to observe the interrelation between the different results, generating new conclusions and predictions, and composing the so-called multilayer model of analysis. It should be noted that the different layers can be both dependent and independent of each other, which can be seen later in this scientific work. There are three defined layers as shown in Fig. 1:

- The initial layer, where the main scientific topics are identified by applying data mining and network analysis techniques in the field of keywords.
- The second layer, where this approach is complemented and made more robust by identifying the most HRI-related terms in the title and abstract fields through natural language processing (NLP) techniques.
- The third layer, where the predictive approach will make it possible to identify the main topics in the near future through techniques based on recurrent neural networks (RNN-LSTM), and the prediction is consolidated with the identification of the most related terms using word embedding techniques.

The multilayered model (comprising the three layers described above), the trend prediction, and the combination of different tools and techniques (bibliometric, data mining, network analysis, NLP and RNN-LSTM), makes the developed methodology innovative in providing a complete and accurate overview of future trends and needs in the field of HRI. Furthermore, by means of network analysis, NLP and RNN-LSTM applied

in the methodology, the aim is, on the one hand, to identify the most important author keywords and their interrelationship by generating co-occurrence matrices and network analyses, in order to identify clusters and the structure of research in the field of HRI. On the other hand, to discover the most HRI-related terms, using word embedding techniques to represent the words in the corpus, as well as to predict future relevant terms. Moreover, the holistic approach of analyzing titles and abstracts separately in all stages and layers of the model, allows for a more comprehensive view of the research topics, as well as identifying relevant terms that might be overlooked in an aggregate analysis.

Finally, in order to provide the reader with the structure followed in this scientific work, the following is a brief description of the different sections developed:

- Methodology, where the methods used, their different combinations and the parameters used, among others, are described in detail.
- Results, where the different results are detailed for each of the three layers, both quantitatively and visually.
- Discussions and conclusions, where the most important results are described together with their usefulness. In addition, it ends with the possible future scientific research with which to continue this article.

# METHODOLOGY

The methodology applied can be seen in Fig. 2, which also shows the steps followed and the stages developed to achieve the objective of this research work, and the different stages can be summarized as follows:

1. First stage: research articles on HRI for the period 2020–2021 are identified in the Scopus database by formulating a specific search query (the first and second stages generate layer 1).
2. Second stage: the dataset is imported into VantagePoint and Gephi to generate co-occurrence matrices and networks that identify author keywords and their relationships (the first and second stages generate layer 1).
3. Third stage: titles and abstracts are cleaned and processed using VBA, and NLP techniques are applied with Word2Vec in Python to generate word embeddings models, identifying relevant terms (this third stage generates layer 2).
4. Fourth stage: an RNN-LSTM in Python is used to train a model with the corpus and generate new abstracts and titles, identifying and quantifying potential terms to highlight in HRI in the short term, and again accompanied by NLP techniques (this fourth stage generates layer 3).

Regarding the three layers described above and corresponding to the multilayer model, these can be seen in different colors in Fig. 3, as well as their dependence/independence with the rest of the layers according to the shared stages and their respective links. As indicated and discussed in more detail below, the methodology developed, in stages and layers, uses different tools depending on the type of analysis to be carried out and the objectives to be achieved.

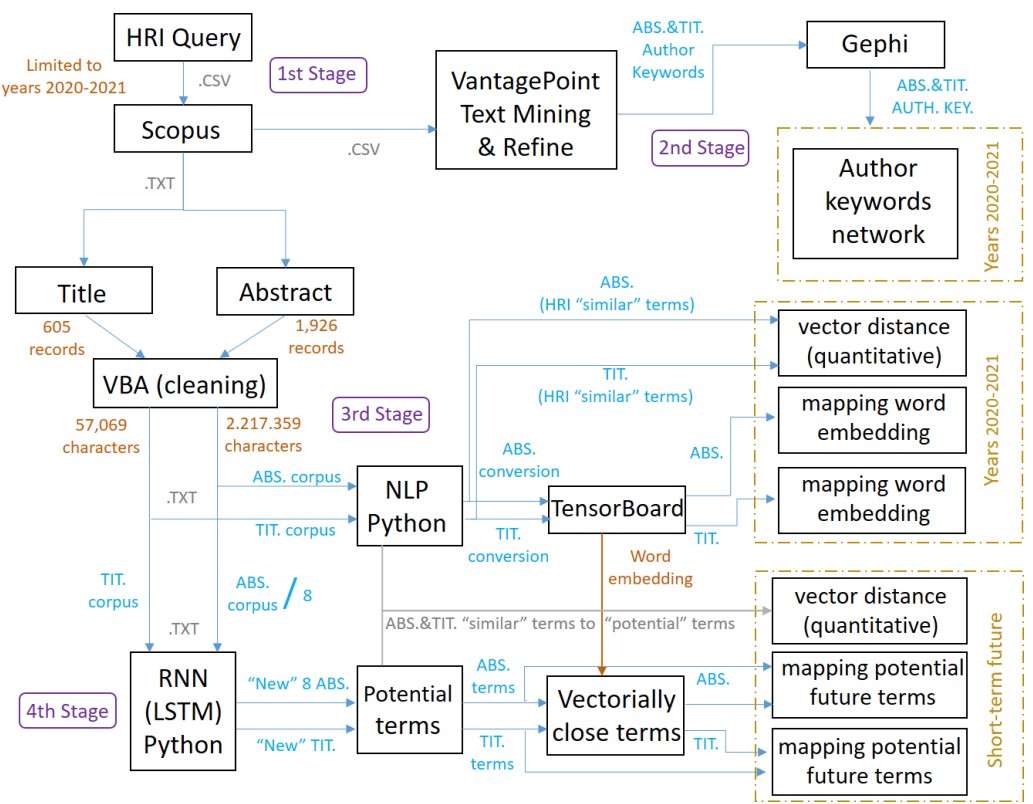

**Figure 2** Scheme referring to the methodology applied and its stages.

## First stage

In the first stage, the research articles of the research topic that will be the dataset of study are identified, as shown in Fig. 4. For this purpose, a search query is formulated, based on the term HRI, and limited to the period 2020–2021. This time limitation is to allow the defined neural network to have a reasonable data processing time, as it has to process less information, thus avoiding problems arising from a high computational load, as detailed at the end of this section. In our scientific work, the total number of hours during which the RNN-LSTM has been working with the applied configuration has been 80 h, divided into different parts. The search is carried out in the scientific database Scopus ("Elsevier, S. Scopus: Content coverage guide.", 2021), one of the largest databases of citations and abstracts of peer-reviewed literature. The queries developed in the HRI field have been the following, avoiding acronyms, as these may refer to other terms in other non-HRI related fields.

• Query referring to author keywords:

*AUTHKEY ("human–robot-interaction") OR AUTHKEY ("human–robot interaction") OR AUTHKEY ("human robot interaction") AUTHKEY ("human–robot-interaction") OR AUTHKEY ("human–robot interaction") OR AUTHKEY ("human robot interaction") AND (LIMIT-TO (PUBYEAR,2021) OR LIMIT-TO (PUBYEAR,2020))*
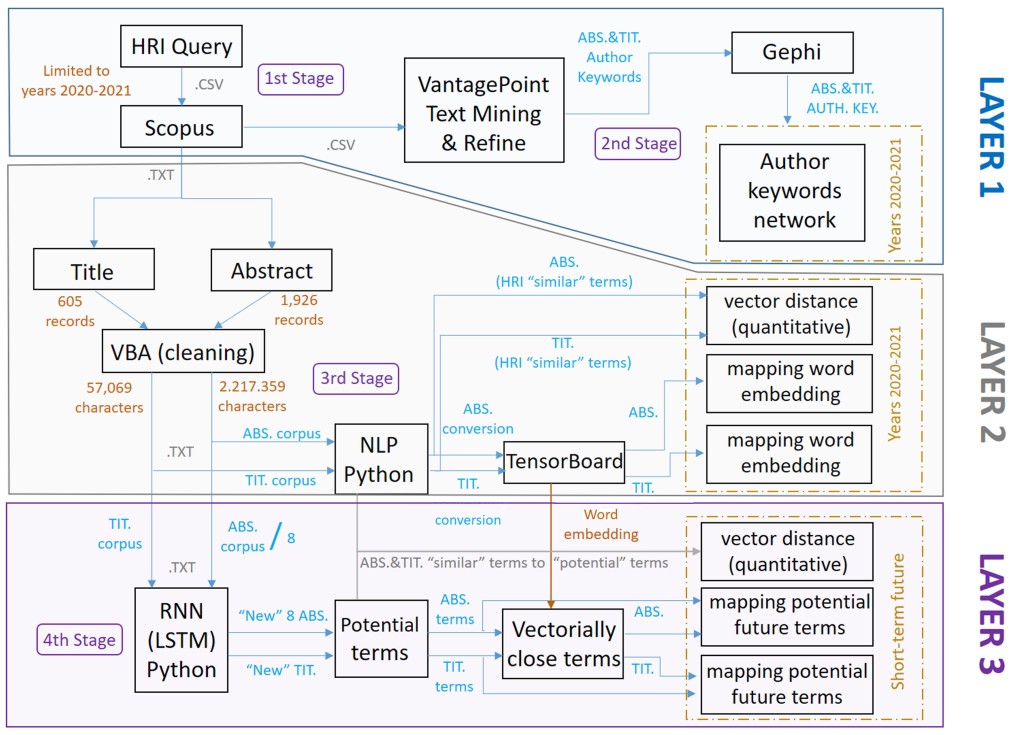

**Figure 3** Scheme referring to the methodology applied and its layers.

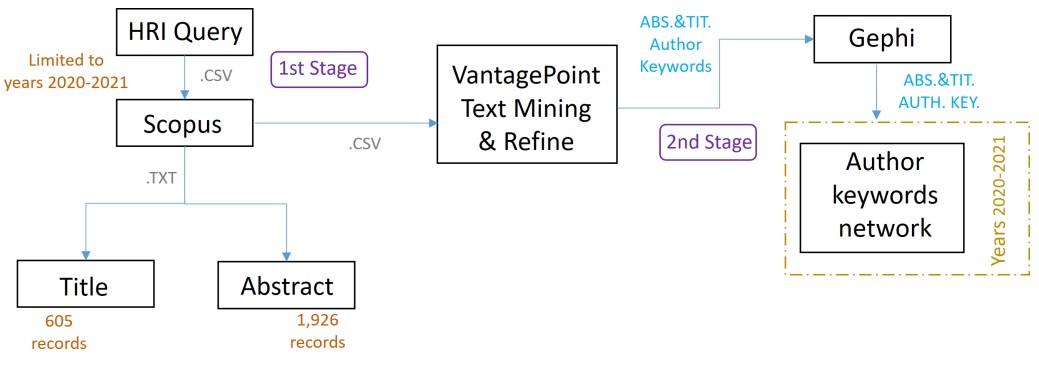

**Figure 4** Scheme referring to the methodology applied in stages 1 and 2.

• Query referring to titles:

*TITLE ("human–robot-interaction") OR TITLE ("human–robot interaction") OR TITLE ("human robot interaction") ABS ("human–robot-interaction") OR ABS ("human–robot interaction") OR ABS ("human robot interaction") AND (LIMIT-TO (PUBYEAR,2021) OR LIMIT-TO (PUBYEAR,2020))*

• Query referring to abstracts:

*ABS ("human–robot-interaction") OR ABS ("human–robot interaction") OR ABS ("human robot interaction") ABS ("human–robot-interaction") OR ABS ("human–robot*

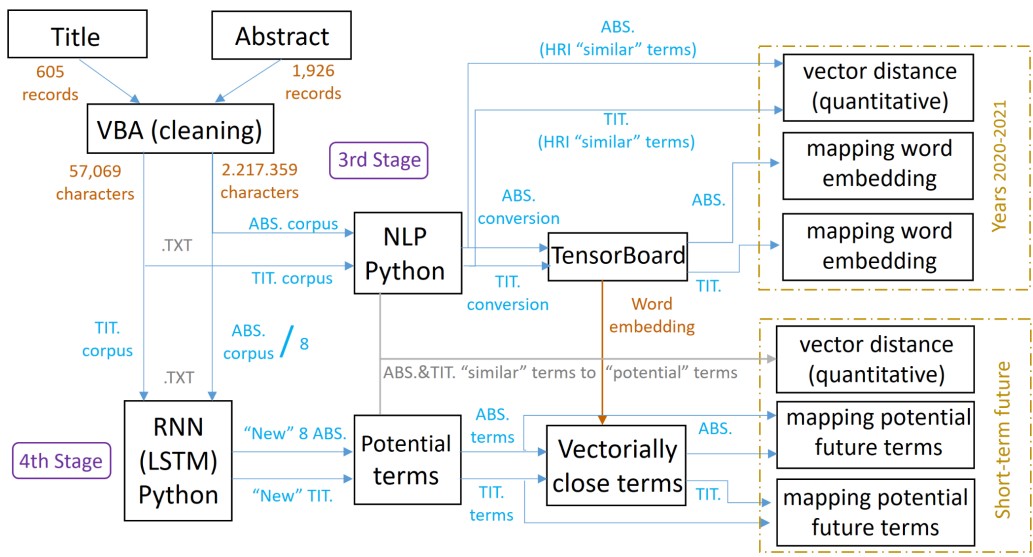

**Figure 5** Scheme referring to the methodology applied in stages 3 and 4.

*interaction") OR ABS ("human robot interaction") AND (LIMIT-TO (PUBYEAR,2021) OR LIMIT-TO (PUBYEAR,2020))*

## Second stage

The second stage begins once the study dataset has been defined, which can also be seen in Fig. 4. In this stage the dataset is imported into the text mining software VantagePoint (VP) and the co-occurrence matrices generated therein, together with the software Gephi, creating a network that allows us to identify the most important author keywords, the relationship between them and clusters.

Both stages 1 and 2 complete layer 1, as can be seen in the figure above, and will be seen later in the results.

## Third stage

Then, in a third stage (see Fig. 5), which completes layer 2, the information obtained corresponding to the abstracts and titles are cleaned by an automated text cleaning process using visual basic for applications (VBA). This automation of the cleaning process using VBA is due, on the one hand, to the large number of abstracts and titles obtained, making manual cleaning unfeasible, as their analysis and cleaning time is very costly and high. And, on the other hand, thanks to the identification of patterns in the structure of the downloaded abstracts and titles, which has facilitated the standardization of the programming code in VBA. Data cleaning is absolutely necessary, since the information obtained not only includes the abstracts and titles, but also embedded information that only adds "noise", which would lead to incorrect training and analysis in the use of NLP and RNN-LSTM

techniques. In this automated process, the following "noisy" information is removed from the abstracts and titles obtained:

- Uniform resource locators (URL) referring to the address of the article in Scopus that accompany the information in the titles and abstracts.
- A "semicolon", which delimits the different titles or abstracts.
- The words "title" and "abstract" at the beginning of the text.
- The symbols (*e.g.,* ©), year of publication, name and city/country of the journal where the article was published, among others.

For this purpose, the unrefined information is pasted into the corresponding tab, depending on whether it is from abstracts or titles, and after clicking on a generated button, it executes the VBA code, copying the abstracts or titles already refined and cleaned, *i.e.,* eliminating the initial noise, into a contiguous tab. In addition, the VBA code itself also takes care of grouping the titles and abstracts separately, so that we can later prepare the corpus with 8-bit unicode transformation format (UTF-8) encoding to be used in Python. The title and abstracts of the scientific articles have been analyzed separately, given that a more complete terminology has been obtained to include information on the theoretical framework, design and application of the research, thus covering the HRI research topics more broadly. The abstract is no more than a summary of a scientific article, in which there may be keywords for our research, and others that are not as important (hence the need for exhaustive cleaning). Consequently, we try to concentrate on the highlights in the titles of the scientific article, each word being relevant. Therefore, separate analysis allows us to focus on the HRI field in more detail without discarding any term related to it. Then, natural language processing (NLP) has been executed and applied. To do so, we have started from the corpus of titles and abstracts generated previously, obtaining a model for each one of them that contains information about the set of feature vectors that represent the words of the corpus (word embeddings). This model provides us with both quantitative and vectorial information on the terms closest to HRI and, therefore, more similar or related to this concept. The vector distances, on the one hand, have been calculated using a Word2Vec (NLP) model executed in Python. The decision to use Python, and not TensorBoard Embedding Projector, is due to the fact that the latter only returns the vector distances of approximately the 8 vectorially closest words referring to a specific term, in our case, HRI. On the other hand, the decimal precision in the vector distances of the words referring to the titles is much higher than with respect to the abstracts, owing to the fact that titles are composed of far fewer records, and therefore characters.

The most important and notable Python configuration used in all this scientific work related to the Word2Vec model corresponding to NLP has been the following:

- Vector size: The dimensionality of the word vectors used was $n = 200$. Moreover, this vector size matches perfectly with the Word2Vec 10K option of the TensorBoard Embedding Projector.
- Window: The maximum distance between the current word and predicted word within a sentence has been set to 5.
- min_count: All words with a total frequency lower than 5 have been ignored.

- hs (hierarchical softmax) = 0 and negative = 10. Since negative (specifying how many "noise words" should be drawn) is greater than 0 and hs is equal to 0, negative sampling will be used for model training.
- sg (skip-gram) = 1. The architecture used for the training algorithm has been skip-gram.
- Epochs: The number of iterations over the corpus was 10.

This model has subsequently been converted using Python to be used through the TensorBoard Embedding Projector. In this way, it was possible to visually identify the terms that are vectorially closest to the HRI concept, *i.e.,* the terms that are most related and complementary to HRI during the period 2020-2021.

In this case, the most notable configuration used concerning the TensorBoard Embedding Projector has been the following:

- Data option: Word2Vec 10K, since this option adjusts the vector size defined in Python programming [dimension(n) = 200].
- Projection type: t-distributed stochastic neighbor embedding (t-SNE) as it showed better results than principal component analysis (PCA) when visualizing clusters, with a perplexity range between 5 and 50 (*Liu et al., 2021*), and with a learning rate of 10. In order to work with a stable system, the number of iterations has exceeded 60,000.
- Search: The distance used has been the cosine distance, since it works well whether the data distribution is balanced or not, and we are mostly dealing with unbalanced data.
- Data points: In studying the abstract, the number of points has been reduced from 4,795 to 1,000 to eliminate noise. Regarding the title, as there are only 196 points, they have not been reduced as there is no noise to distort the visualization.

**Fourth stage**

Finally, in the fourth stage, which completes layer 3, using long short-term memory (LSTM) recurrent neural network (RNN) in Python, creates a model trained with the corpus, capable of learning the style of the text and generating new sentences and/or abstracts and titles, also schematically described in Fig. 5. LSTM, unlike the RNN, is able to learn from historical or much earlier data, thereby identifying, remembering and memorizing features of long-term historical data and capable of making more accurate short-term predictions (*Xiangxue, Lunhui & Kaixun, 2019*; *Gao & Hong, 2019*; *Rafi et al., 2021*). The different words that compose the new text, together with the frequency in which they appear, make them worthy of consideration as potential terms to be highlighted in the short-term within the HRI theme, so they can be mapped through the word embedding model generated previously and thus analyzed. In addition, these potential terms are also appraised quantitatively through the Word2Vec (NLP) model previously executed in Python to observe the vector distance (closeness) that these potential terms share with other words, the latter being the terms that "by affinity" may eventually accompany the terms classified as potential terms to be highlighted in the short-term. This makes it possible to give greater robustness to the items identified in the two previous stages and to define new paths of research.

The optimization algorithm used in Python in reference to the RNN with LSTM architecture was Adam's algorithm, and this, in turn, was accompanied by the following configuration with respect to the most important hyperparameters:

- The Cross Entropy mathematical function used to calculate loss function.
- The batch size used was 128, together with 20 epochs.
- The activation function used was Softmax.

It is important to emphasize that because the abstract corpus is composed of more than 1.9 million characters, it has been divided into 8 parts of similar numbers of characters in order to reduce any possible problems due to the high computational load involved in its execution.

## RESULTS

In order to present the results, it is useful to recall which tools, techniques and methods are used in the different layers that make up the multilayer model.

- VantagePoint: text analysis and data mining tool.
- Gephi: graph visualization and analysis tool.
- TensorBoard Embedding Projector: visualization tool for data embedding.
- VBA (Visual Basic for Applications): programming language for automating tasks in Microsoft Office applications.
- NLP Word2Vec: natural language processing technique for creating vector representations of words.
- RNN with LSTM: machine learning method for modeling temporal sequences.

Regarding the hyperparameters used, these have been detailed in the methodological section, the unique "parameter" they share being temporal limitation to the years 2020-2021 applied in the search query.

### Layer 1: mapping author's keywords around HRI

To produce the first of the layers, and thus obtain the first results for further analysis, an author's keyword co-occurrence network has been generated. A co-occurrence matrix is a mathematical representation that shows the frequency with which two elements, words or terms, appear together in a data set. The analysis of the network not only permits visualization of the co-occurrence matrix based on the authors' keywords, but also their quantitative analysis. In this way, the network can be used to visualize the terms with the highest number of co-occurrences represented by the node size (weighted degree) and, in turn, the modularity of the network which groups or clusters terms that identify related research fields (see Fig. 6). In addition, the analysis of the network based on the network diameter makes it possible to identify the terms with the highest level of centrality (betweenness centrality) and the most influential ones (closeness centrality).

The author's keywords that have appeared most often and, therefore, with the greatest weight (and node) are concepts such as social robots (*Ulhøi, 2022*), physical human–robot interaction (pHRI), social robotics, human–robot collaboration (HRC), artificial

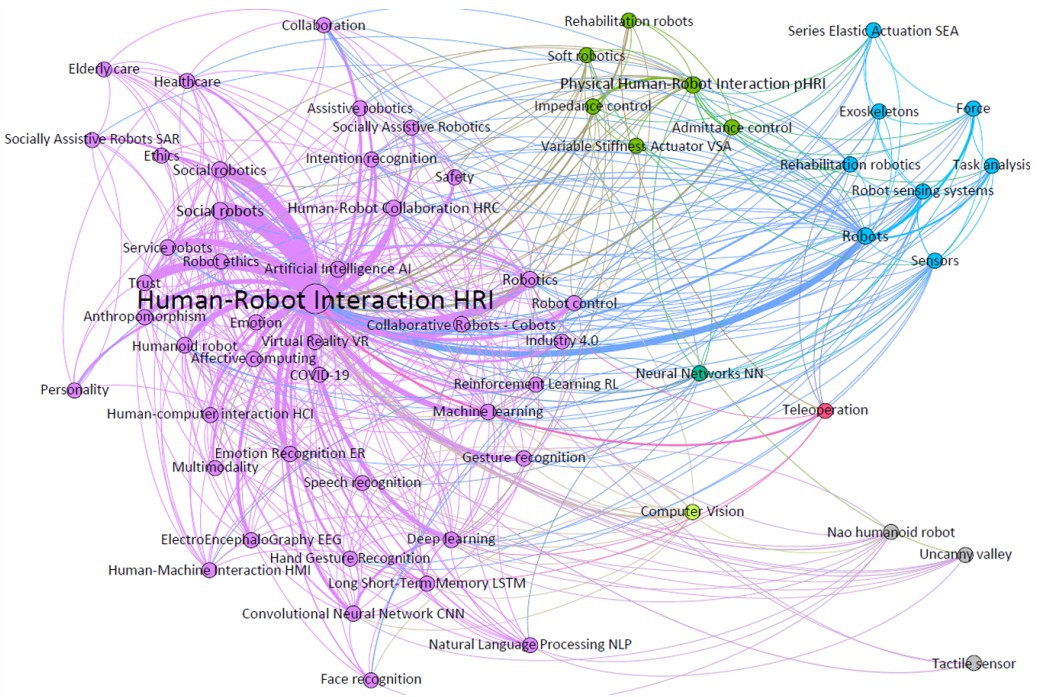

**Figure 6** Author's keywords co-occurrence network.

intelligence (AI) or collaborative robots (cobots), among others, and all of them referring to the main cluster (purple), except pHRI. Regarding the rest of the highlighted clusters: the main concept of the green cluster is the term pHRI and its relation to the terms impedance and admittance control (*Ott, Mukherjee & Nakamura, 2010*); the blue cluster with the word robot as the main concept and its strong relation to HRI; and finally, the small pink cluster with teleoperation (*You, Zhou & Ding, 2023*) as main term and with a close relation to HRI.

As shown in Table 1, the network identifies that the most used, best intermediated and most influential topics around HRI are those linked to research fields around robotics (human–robot collaboration (HRC), physical human–robot interaction (pHRI), social robotics, robotics…); industrial or domestic applications (collaborative robots (cobots), humanoid robot, service robot and social robots), and to computational algorithms or applications such as machine learning (ML), deep learning (DL) and artificial intelligence (AI).

## Layer 2: identifying topics closely related to HRI

To identify the terms with greater affinity related to HRI in reference to the abstracts, the vector distances of various terms were obtained quantitatively by applying Word2Vec (NLP) models generated in Python, as shown in Table 2, with 0 being the point or word vectorially closest to the HRI concept and 1 the farthest. The results describe general

**Table 1  Ranking of keywords with larger relationships, intermediarity and influence.**

| Ranked | Most relationships keywords | Weighted degree | Intermediary keywords | Betweenness centrality | Influence keywords | Closeness centrality |
|---|---|---|---|---|---|---|
| 1 | Human–robot interaction HRI | 4353.0 | Human–robot interaction HRI | 5847823.84 | Human–robot interaction HRI | 0.64726 |
| 2 | Social robots | 556.0 | Physical human–robot interaction pHRI | 526243.98 | Robotics | 0.46122 |
| 3 | Physical human–robot interaction pHRI | 471.0 | Social robots | 328967.35 | Robots | 0.45885 |
| 4 | Social robotics | 326.0 | Robots | 281897.67 | Social robots | 0.45531 |
| 5 | Robotics | 333.0 | Robotics | 256821.12 | Physical human–robot interaction pHRI | 0.45351 |
| 6 | Human–robot collaboration HRC | 285.0 | Human–robot collaboration HRC | 232673.04 | Collaborative robots - Cobots | 0.45188 |
| 7 | Robots | 420.0 | Deep learning | 201376.18 | Machine learning | 0.45112 |
| 8 | Deep learning | 265.0 | Humanoid robot | 175214.08 | Social robotics | 0.45056 |
| 9 | Artificial intelligence AI | 260.0 | Social robotics | 167275.86 | Human–robot collaboration HRC | 0.44930 |
| 10 | Collaborative robots - Cobots | 236.0 | Collaborative robots - Cobots | 149113.83 | Humanoid robot | 0.44616 |

**Table 2  Ranking of words related to HRI obtained from the abstract corpus.**

| Ranking of related words | Term | Vectorial distance |
|---|---|---|
| 1 | human–robot | 0.262 |
| 2 | collaboration | 0.267 |
| 3 | interaction | 0.270 |
| 4 | human–computer | 0.279 |
| 5 | human–machine | 0.296 |
| 6 | successful | 0.309 |
| 7 | HCI (human–computer interaction) | 0.312 |
| 8 | human–human | 0.318 |
| 9 | guidelines | 0.336 |
| 10 | studying | 0.338 |

performance fields such as relationships (human–robot; human–computer; human-machine), research methodologies (guidelines, study) and machine skills and results (collaboration, interaction, successful).

Subsequently, by converting to a tabular structure in Python, it has been possible to import the information entailed by the vector distances and metadata in the TensorBoard Embedding Projector, thus obtaining a visual way to depict the different word embeddings between the period 2020–2021 (Fig. 7). In this case, as in the quantitative analysis, the higher the affinity with respect to the HRI concept, the smaller the distance or the closer the proximity. For this purpose, Fig. 7 has been divided into two images, the right image (Fig. 7B) being a magnification of the image on the left (Fig. 7A).

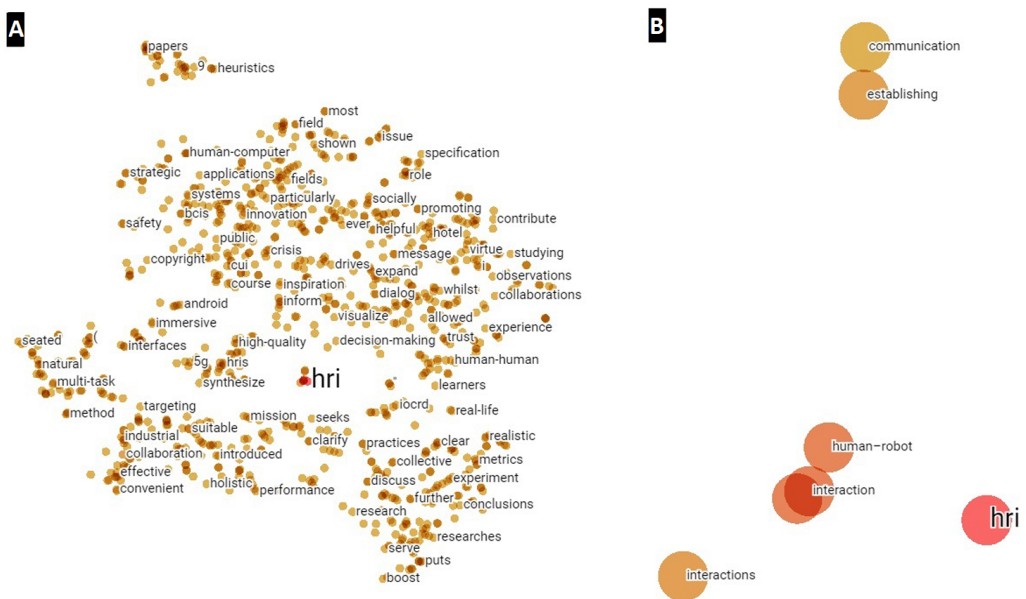

**Figure 7** **Word embeddings of the term HRI created from abstracts for the year 2020–2021.** (A) Main network for visualizing possible clusters. (B) Zoom around HRI.

Finally, it should be noted that no significant clusters are observed for the abstracts, despite having reduced the number of points from 4,795 to 1,000 in order to eliminate the existing noise.

With respect to the titles, the steps to follow are the same as those described above for the abstracts, *i.e.,* by converting to a tabular structure in Python, the information is imported into the TensorBoard Embedding Projector and a visual analysis can be made. In this case, since there are only 196 points and no noise generated, no point reduction was necessary to visualize clusters around the HRI concept, as shown in point 1 of Fig. 8. In Fig. 8, different zooms have also been performed with respect to the previous numbering, indicated by Figs. 8B, 8C & 8D. In this way, as with the abstracts analyzed above, the terms closest to HRI are identified. These affinities and identification of terms, discussed later in this scientific work, will serve as a basis for generating the multilayer analysis model.

As with the abstracts, a quantitative analysis is carried out to observe, strengthen, verify and give veracity to the results obtained visually. Table 3 quantitatively ranks the vector distances of various terms of the titles with respect to the HRI concept, continuing with the standard of 0 as the closest point or word vectorially, and 1 as the farthest.

## Layer 3: predicting potential short-term terminology regarding HRI

In this third layer, we have used the corpus of abstracts and titles previously obtained and refined, to train a long short-term memory (LSTM) recurrent neural network (RNN) using Python. This RNN is defined in more detail as a "generative" LSTM, since once it has been trained and the learning process has been carried out, it is capable of generating new text. In our case, we have achieved the automatic generation of eight new abstracts and a title,

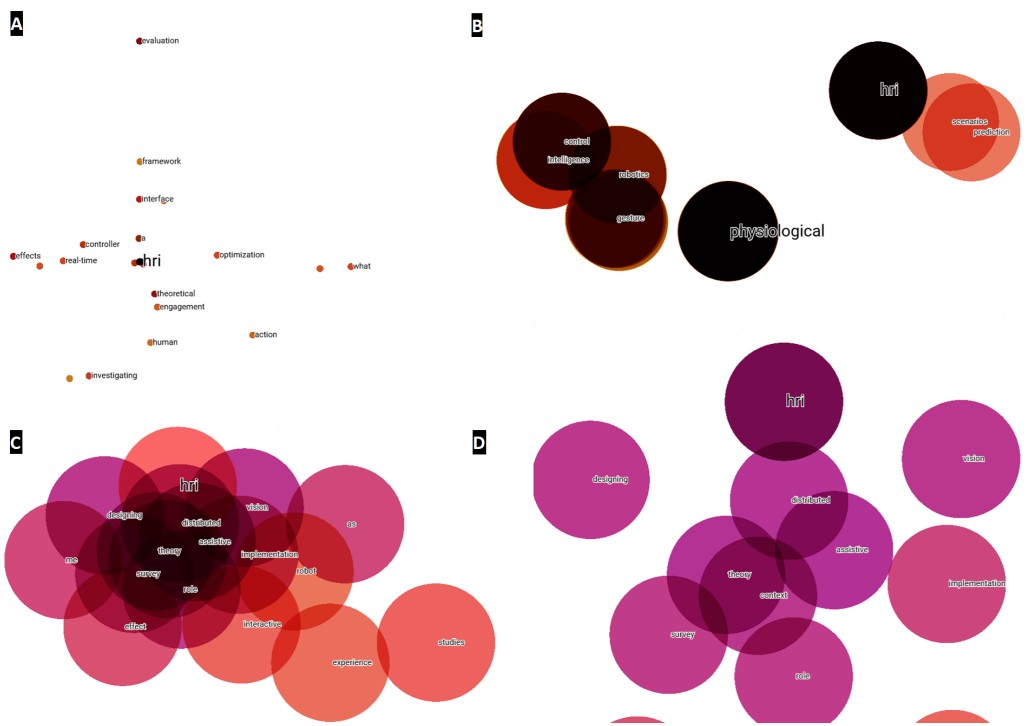

**Figure 8** **Word embeddings of the term HRI created from titles for the year 2020–2021.** (A) Main network for visualizing possible clusters. (B) First zoom level around HRI. (C) Second zoom level around HRI. (D) Third zoom level around HRI.

**Table 3** **Ranking of words related to HRI obtained from the title corpus.**

| Ranking of related words | Term | Vectorial distance |
| --- | --- | --- |
| 1 | anthropomorphism | 0.000629 |
| 2 | vision | 0.000690 |
| 3 | distributed | 0.000723 |
| 4 | intelligence | 0.000740 |
| 5 | assistive | 0.000749 |
| 6 | designing | 0.000750 |
| 7 | theory | 0.000779 |
| 8 | environment | 0.000786 |
| 9 | movements | 0.000786 |
| 10 | robotics | 0.000793 |

with which to identify potential terms to be highlighted and studied in the short-term by the scientific community with respect to the concept or field of HRI.

Figure 9 shows the three steps followed to select terms considered as potential.

- In the first step, the most repeated terms or with the highest frequency of occurrence among the total set of abstracts and titles.

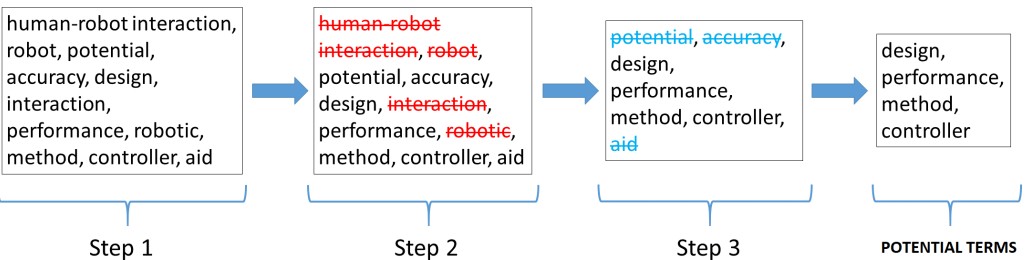

**Figure 9** Criteria followed to select potential terms with respect to the HRI field.

- Subsequently, once a list of the most repeated terms was obtained, in the second step these terms were refined by eliminating those terms that intrinsically include words corresponding to the HRI concept, namely human–robot interaction, robot, interaction and robotic.
- Finally, with the terms selected in the previous step, and to make a more related and precise selection to the HRI term, in this third and final step, stage 3 of the methodology is performed again (NLP technique), searching for step 2 terms in both the corpus of abstracts and the corpus of titles. Only those terms that appear in both corpora are selected as potential terms.

Once these three steps have been applied, the words 'design, performance, method and controller' can be seen to be the terms with the greatest potential, focus and growth for HRI research to be developed in the short-term. This statement is based, on the one hand, on the fact that during the LSTM-RNN takes the frequency of occurrence of the different terms, among other hyperparameters, into consideration during its learning process and subsequent text generation process. If a term has a high frequency, it is synonymous with a keyword (*Wang & Ning, 2020*), and is therefore a hot topic in the scientific community. On the other hand, the fact that an author of a scientific paper uses these terms in the title of the article or in its abstract or in both, denotes a desire to highlight them due to their significance and importance within a specific scientific field or area of study. In addition, titles and abstracts are the most important parts of a research paper, which include the most notable words and terms (*Tullu, 2019*). Therefore, by taking the word embedding analysis of titles and abstracts through the TensorBoard Embedding Projector as a starting point, Figs. 10 and 11 visually represent the existing affinity between the potential terms to HRI and the words vectorially closest to these.

Before visually analyzing the word embedding from titles and abstracts through the TensorBoard Embedding Projector, and given that the four potential terms will appear several times both in the results and in the discussion and conclusions of this scientific work, it is useful to detail the definition and importance of each one of them:

- Design
  - Definition: refers to the planning, development and creation of prototypes and robots and their interfaces to facilitate effective interaction with humans.

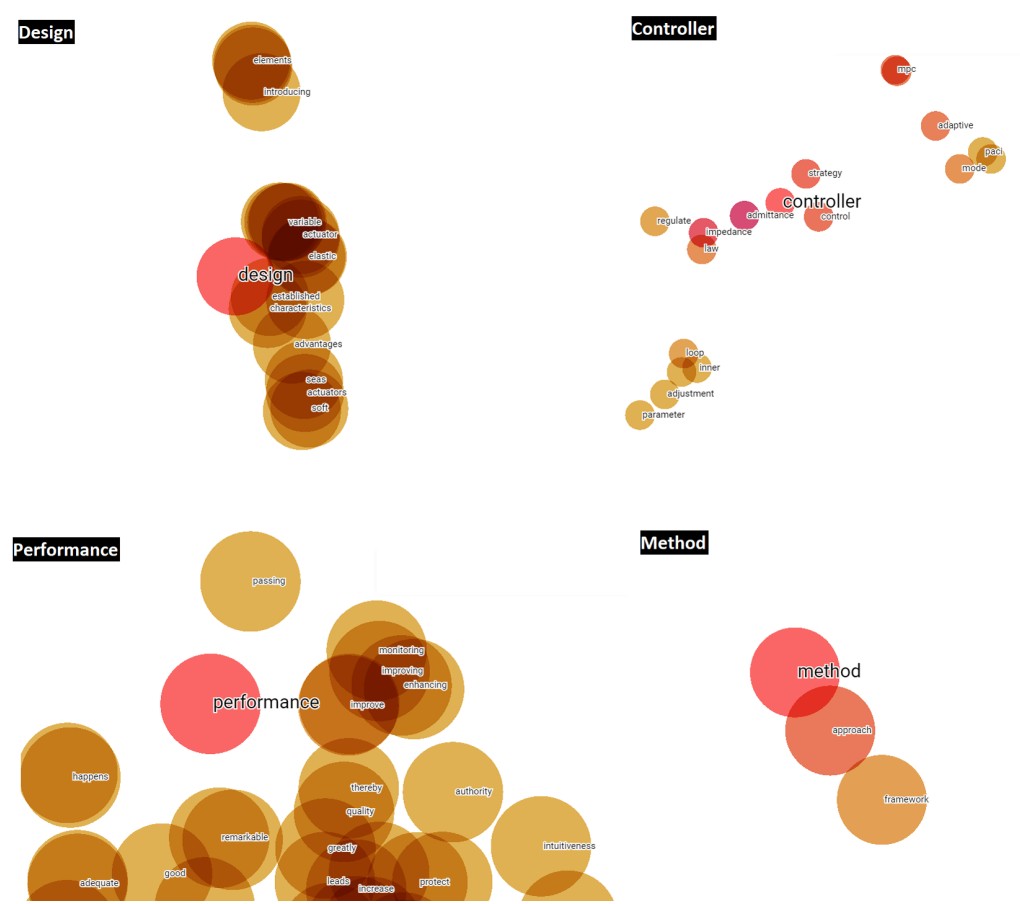

**Figure 10** Words vectorially closer to potential future terms in relation to the abstracts.

 ○ Importance: good design should consider ergonomics, usability, accessibility and the ability of users to understand, interact and communicate with the robot intuitively.
- Performance
  - Definition: refers to the efficiency and effectiveness with which a robot performs its tasks, as well as the quality of human–robot interaction.
  - Importance: evaluating performance involves measuring accuracy, speed, reliability, robustness and user satisfaction during interaction with the robot.
- Method
  - Definition: techniques and procedures used to investigate, develop and evaluate human–robot interactions.
  - Importance: methods may include user studies, controlled experiments, use and optimization of algorithms and models, simulations, and qualitative and quantitative analyses to better understand how to improve HRI.
- Controller
  - Definition: refers to the system or algorithm that regulates the robot's mechanical behavior to perform specific tasks and respond appropriately to human actions.

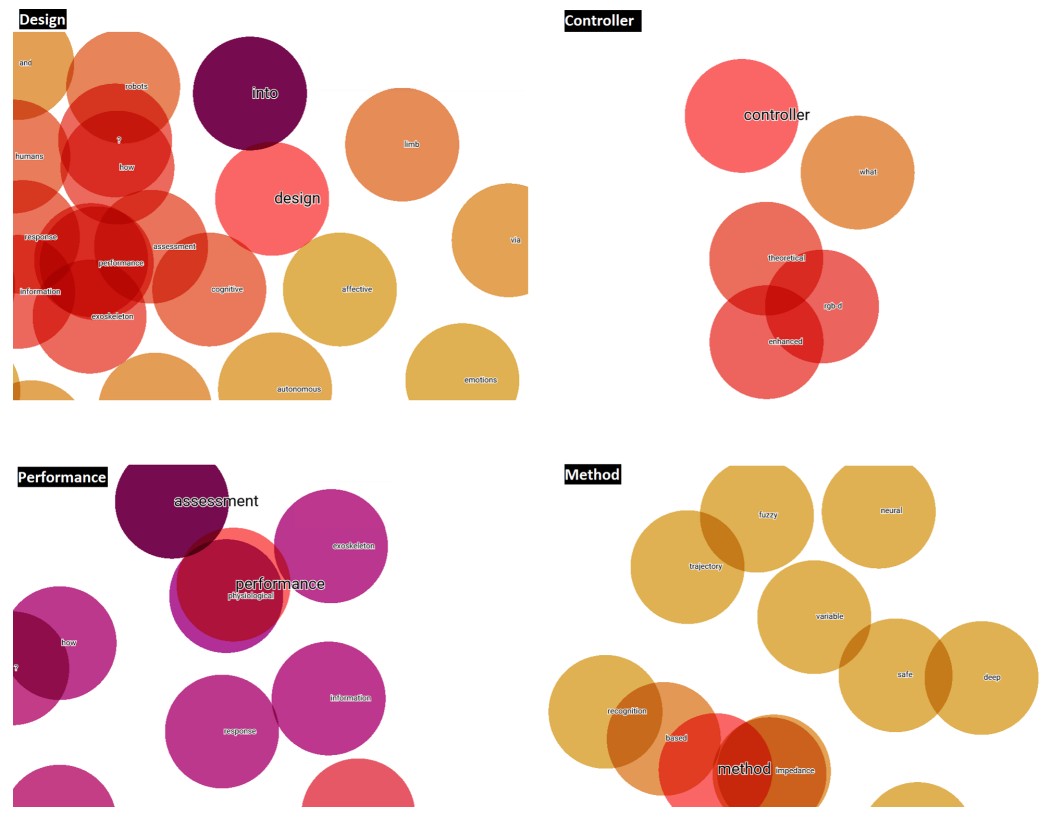

**Figure 11   Words vectorially closer to potential future terms in relation to the titles.**

○ Importance: an effective controller enables the robot to act in a safe, predictable and adaptive manner, adjusting its behavior in real time according to the user's needs and actions.

Then, taking the word embedding analysis of titles and abstracts through the TensorBoard Embedding Projector as a starting point, Figs. 10 and 11 visually represent the existing affinity between the potential terms to HRI and the words vectorially closest to these.

In addition, Word2Vec (NLP) models generated in Python are applied to analyze the defined potential terms. As shown in Table 4, the words vectorially closest to the potential terms have been quantitatively calculated, emphasizing that they are terms relating to methods linked to the items or qualities of performance of such.

The definition of the research agenda is based on the three layers obtained (see Table 5). The identified topics have been classified in seven fields: research area, industrial or domestic application, systems (mechanical or program/software), computer algorithms or applications, methods and results, skills or characteristics and others. This classification was made by analyzing the terms obtained and consulting a group comprising two experts in the fields of robotics and ergonomics.

**Table 4   Words vectorially closer to potential future terms [TOP-5].**

| Potential Term | Abstract | Title |
|---|---|---|
| Design | principles [0.352] | communication [0.000951] |
| | guidelines [0.365] | engagement [0.000986] |
| | prototyping [0.370] | enhanced [0.000992] |
| | detailed [0.392] | movements [0.001010] |
| | elements [0.393] | distributed [0.001023] |
| Performance | improves [0.252] | action [0.000697] |
| | robustness [0.257] | case [0.000734] |
| | efficiency [0.259] | human–robot [0.000735] |
| | stability [0.294] | elastic [0.000736] |
| | terms [0.300] | interactive [0.000761] |
| Method | approach [0.230] | controller [0.000864] |
| | proposed [0.235] | admittance [0.000942] |
| | algorithm [0.255] | model [0.000988] |
| | scheme [0.259] | force [0.001004] |
| | optimization [0.275] | based [0.001009] |
| Controller | admittance [0.071] | signals [0.000668] |
| | impedance [0.086] | prediction [0.000714] |
| | scheme [0.143] | modeling [0.000742] |
| | strategy [0.143] | body [0.000762] |
| | loop [0.147] | environments [0.000775] |

The analysis of the most influential keywords related to the research works in the HRI area indicates the important relationship between different sub-fields of study, areas or concepts, such as general robotics with social robotics, physical human–robot interaction (pHRI) and human–robot collaboration (HRC). In turn, with industrial or domestic applications of robotics, such as collaborative robots or cobots, humanoid robot, social robot, robot and service robot, and the essential computational algorithms in the development path such as machine learning (ML), deep learning (DL), neural network (NN), convolutional neural network (CNN) and long short-term memory (LSTM), and their application in artificial intelligence (AI). In addition, control mechanisms such as detection systems are important, plus a characteristic that robots must have is identified as "trust". This analysis is completed with the identification of new terms by analyzing the abstract and titles through natural language processing (NLP). The fields of study are extended to human–computer and human-machine interactions (HMI), and a new area of study linked to virtual-augmented and mixed reality is introduced. Computational systems include the brain-computer interface, and within the category of mechanical systems, vision and motion become important. This analysis allows the most important skills or characteristics that are researched in the area to be addressed with greater definition, such as assistive, wearable, intelligent, interaction, collaboration and communication. In addition, it identifies the most related research methodologies with the greatest growth and development and advancement trends, such as design and theoretical studies, among others.

**Table 5  Research agenda by layering research topics.**

| | | Research area | (Industrial or domestic) applications | Systems (mechanical or program) | Computer algorithms or applications | Methods and results | Skills or Characteristics | Others |
|---|---|---|---|---|---|---|---|---|
| **(a) Layer 1: Extracting Research Topics** | | | | | | | | |
| **Layer 1: Extracting research topics** | Keywords by closeness centrality | Human–robot interaction HRI Robotics Physical human–robot interaction pHRI Social robotics Human–robot collaboration HRC | Robots Social robots Collaborative robots - Cobots Humanoid robot Service robots | Robot sensing systems Robot control | Machine learning Deep learning Artificial intelligence AI Neural networks NN Convolutional neural network CNN Long short-term memory LSTM | | Trust | Sensors (devices) |
| **(b) Layer 2: Completing Research Topics by NLP** | | | | | | | | |
| **Layer 2: Completing research topics by NLP** | Abstracts (Theoretical Framework) | human–robot human–computer human-machine HCI (human–computer interaction) human–human HMI (human–machine interaction) VAM-HRI (Virtual, Augmented, and Mixed-Reality) HRI HHI (human-to-human interaction) | | | brain-computer | successful guidelines studying methodologies workshop | collaboration interaction | community contribute principles interfaces |
| | Title (Objective) | Environment Robotics | | vision movements tasks | | designing theory evaluation exploring investigating theoretical studies | distributed intelligence assistive communication wearable | anthropomorphism use RGB-D (sensors) |
| **(c) Layer 3. Predicting Research Topics by LSTM** | | | | | | | | |
| **DESIGN** | Abstracts (Theoretical Framework) | | | elements | | principles guidelines prototyping innovative integration development implementation methodology theoretical | | detailed describe provides concept alternative considerations practical conclude outline explore |
| | Title (Objective) | | | | | implementation | communication movements assistive interactive mobile elastic | engagement enhanced distributed signals skin features role technologies context environments scenarios older |
| **PERFORMANCE** | Abstracts (Theoretical Framework) | | | | | improves improve feasibility | robustness efficiency stability accuracy effectiveness agility | terms comparable improvement proved demonstrated superior transparency confirm convergence verify efficacy |
| | Title (Objective) | environment factors technologies | human–robot | | | action case theory | elastic interactive cognitive gaze assistive wearable communication | limb effect engagement use response RGB-D (sensors) |
| **METHOD** | Abstracts (Theoretical Framework) | | | | | algorithm optimization analytical experimentally model-based framework genetic feasibility logic | convergence | approach proposed scheme outperforms superior demonstrated validate verify proposal proved |

*(continued on next page)*

**Table 5** (*continued*)

| | | Research area | (Industrial or domestic) | Systems (mechanical | Computer algorithms | Methods and results | Skills or Characteristics | Others |
|---|---|---|---|---|---|---|---|---|
| | Title (Objective) | | | trajectory control impedance | | model fuzzy modelling interaction implementation prediction planning survey | controller admittance dynamic mobile smart | force based tracking signals |
| CONTROLLER | Abstracts (Theoretical Framework) | | pHRI | VSA (variable stiffness actuator) PD (proportional derivative) controller | | scheme strategy Lyapunov (stability in motion) MPC (model predictive control) | admittance impedance loop adaptive predictive switching disturbance nonlinear | mode observer law Cartesian |
| | Title (Objective) | | | | | prediction modelling survey implementation trajectory fuzzy application interaction | gaze mobile | signals body environments planning arm distributed enhanced RGB-D (sensors) |

The first two layers provide the information necessary to determine what the main HRI related development carried out in science has been. In the third layer, the results obtained allow us to predict the four most important short-term terms related to the scientific development of HRI, and at the same time, to identify the topics most related to them. The most notable results of these main terms describe skills of those robots linked to the area of study, *i.e.,* the term DESIGN is accompanied by a method based on prototyping, development and implementation, and skills aimed at communication, movement, elastic, interactive and assistant. The term PERFORMANCE is characterized by improvement actions, and skills aimed at robustness, stability, accuracy, elasticity, cognitive and wearable. The term METHOD highlights the quality of convergence, and specifies the main methods linked to scientific developments: optimization, algorithms, model-based, prediction, among others. Finally, the term CONTROLLER is linked to two terms associated with mechanics, variable stiffness actuator (VSA) and proportional derivative (PD) controller, and terms associated with predictive and stability methods are highlighted, with skills associated with controllers such as switching, adaptive, admittance, impedance, *etc.*

## DISCUSSION & CONCLUSIONS

The results of the study allow us to conclude that the methodology followed to define the research agenda is correct and provides a solid and consolidated basis for the analysis of topics based on bibliometric techniques.

The first layer of the research agenda, directly linked to scientific development carried out in the last two years, highlights the importance of the intensity of HRI application, not only in the industrial world but also in the social and service world, and the main characteristic of this human–robot interaction is trust. All this achieved on the basis of advanced computational algorithms such as machine learning (ML), deep learning (DL) and neural networks (NN).

The second layer complements the previous one, shifting scientific developments towards objectives, qualities or actions that have to perform these interactions: vision, sensors, communication, collaboration (relevant actions within the definition of HRI), assistive, and above all wearable, jargon closely associated with small computers such as smartwatches, as well as the importance of the approach to the human from an anthropomorphic-based perspective.

Once the vision of the technological development carried out has been defined, the third layer makes it possible to define the most important terms expected in the near future. Design, performance, method and controller become the key references, therefore human–robot interactions become open ended in the design of elements that allow skills related to movement, interaction, communication, assistance and mobility to be improved. The performance of human–robot interactions should be robust, stable, efficient, effective, accurate, interactive, cognitive, lifelike gaze, wearable and agile. The methods for conducting research acquire their relevance, highlighting modeling, prediction, implementation, algorithms, fuzzy logics, optimization, experimentation and analytics. Finally, in terms of interaction controllers, physical human–robot interactions set the field of research, highlighting the methods or Lyapunov's stability theory, which mathematically seeks to obtain optimal behavior in a robot's control systems (*Zhang et al., 2018*). Within the interaction controllers, it is also worth highlighting the predictive control models, as well as the characteristics or types of controllers such as admittance, impedance, switching, disturbance, loop, proportional derivative controller and variable stiffness actuator.

Consequently, the research proposes a future agenda of relevant actions linked to the skills or characteristics of human–robot relationships that will allow research groups to define their strategies in a more focused fashion.

Therefore, it can be concluded that this field of research is a very active and constantly evolving, so predicting future trends and needs opens the way to new in-depth research related to the HRI concept and collaborative robotics, thus achieving an increased generation and transmission of knowledge for the scientific community. In addition, this academic work fills a gap in scientific literature by pioneering the development of a research agenda in the field of HRI, with the methodology developed and the results obtained being its greatest added value.

Although there is no research agenda in the field of HRI and collaborative robotics, so no comparison with other studies or similar models can be made, as indicated at the beginning of this section, the methodology applied is correct, and the results obtained are accurate:

- In the first layer, by means of data mining, it has been determined that one of the main characteristics of this human–robot interaction is trust, an identical conclusion reached by *Mohammad & Nishida (2016)* when applying data mining in his scientific work. These conclusions are also reached by various authors using different methodologies, as in the works of *St-Onge, Reeves & Petkova (2017)* and *Billings et al., (2012)*.
- In the second layer, using NLP techniques, it has been determined that vision, sensors, communication and collaboration are relevant actions within the definition of HRI.

These same conclusions are reached by the authors *Li, Bill & Bringardner (2023)* (vision and sensors) and *Siwach & Li (2024)* (communication and collaboration), when applying NLP techniques in their scientific works. These conclusions are also reached by other authors using different methodologies, as in the work of *Tashtoush et al. (2021)*.

- In the third and final layer, by means of RNN-LSTM, it has been determined that design, performance, method and controller become the key future references of HRI. These potential terms can also be reflected in the work of *Ren & Xi (2022)* where they apply RNN methods.

Finally, as the study is based on academic literature, the analysis is exclusively scientific. Therefore, in order to cover the HRI concept and collaborative robotics in more depth, and to complement this article, a proposed future step would be to carry out a technological analysis, with which to similarly develop a technological development agenda. For this, patents would be used as a starting point, as they contain more than 90% of the recent technological information existing in the world, as they are a repository of knowledge derived from research activities (*Zha & Chen, 2010*; *Arunagiri & Mathew, 2017*). Within the different sections available in a patent, claims are considered fundamental parts of technological information extraction, as they describe the technological innovations that are intended to be protected (*Qiu & Wang, 2021*; *Fujii, Iwayama & Kando, 2007*). Therefore, this may be considered a suitable starting point for technology information extraction, in order to conduct a more comprehensive and complete analysis of the field of HRI and collaborative robotics in the future.

### Funding
The authors received no funding for this work.

### Competing Interests
The authors declare there are no competing interests.

### Author Contributions
- Jon Borregan-Alvarado conceived and designed the experiments, performed the experiments, analyzed the data, performed the computation work, prepared figures and/or tables, authored or reviewed drafts of the article, and approved the final draft.
- Izaskun Alvarez-Meaza performed the experiments, analyzed the data, prepared figures and/or tables, authored or reviewed drafts of the article, and approved the final draft.
- Ernesto Cilleruelo-Carrasco conceived and designed the experiments, authored or reviewed drafts of the article, and approved the final draft.
- Rosa Maria Rio-Belver performed the computation work, authored or reviewed drafts of the article, and approved the final draft.

### Data Availability
The raw data and code are available in the Supplemental File.

## Supplemental Information

Supplemental information for this article can be found online at http://dx.doi.org/10.7717/peerj-cs.2335#supplemental-information.

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
