# Peer review of "Human–robot interaction: predicting research agenda by long short-term memory"

_PeerJ Computer Science, doi:10.7717/peerj-cs.2335_

## Round 0.1 · original submission · Major Revisions

· Academic Editor

Major Revisions

After carefully considering the reviews and assessing your manuscript, I would like to invite you to revise and resubmit your manuscript for further consideration. The reviewers have provided constructive comments that will help strengthen your work. Please address each of these points thoroughly in your revised manuscript. Additionally, ensure that you provide a detailed response letter outlining how you have addressed each comment raised by the reviewers. This will help the reviewers and myself to evaluate the changes made to the manuscript.

Reviewer 1 ·

Basic reporting

The manuscript lacks proper English, so it is highly advised to have the proofreading completed by professional services or fluent English support speakers.
The Literature Review should introduce the State of Art Section in an organized way.
The figures submitted by the authors are not precise and per the Journal Guidelines.
The figures should be modified for clarity of vision and understanding into vector graphics.
Several terms in the Formal results do not include the definitions used. It is highly recommended that you update these terms and their meanings.

Experimental design

The paper entitled Human-robot Interaction: Predicting Research Agenda by Long Short-term Memory produces a review of the HRI techniques published worldwide to enable industry 5.0 level techniques. The article identifies the main research topics through network analysis, natural language processing (NLP), and word-embedding techniques.
The topic seems interesting, as several modifications are needed in the experimental design.
The SCOPUS Abstract Collection dataset comprises 1949 entries till 2021 (the latest one). The authors must inculcate several more entries in the system from Scopus to make the paper better and stronger.
Among the 612 entries of the SCOPUS Title Raw Dataset, more recent advanced-level titles from the near past (which are found missing) are likely to be included.
Only Industry-level titles and abstracts are selected from the corpus data, which might result in the loss of several HRI-based articles. In this case, it is advised to modify the wild card search.
The VBA Code seems to be fine, but the authors are expected to submit the VBA macro-enabled file in the legends for a clear understanding of how the code works and an experimental demonstration.
A clear understanding of Corpus cleaning in the Title and abstract clean file is missing from the manuscript. The authors must explain the procedure clearly.

Validity of the findings

The Corpus data was also extracted in 2021, which seems outdated. To produce better results, the authors should regenerate the data and reiterate the findings.
To understand them better, the Word2Vec results should be evaluated using various epochs and other parameters. This will strengthen the potential findings.
The authors should clearly mention the strategy behind selecting the Seed value in the LSTM Code ipynb file.
Several more potential keywords should be identified from the various analysis techniques the authors tried to include in this study.

Additional comments

The abstract should be re-written, mentioning the complete scenario and methodology adopted by the authors in the manuscript.
Several new citations and references should be included in the manuscript.
Kindly ensure the Table Citations and Figure Referencing appear in an ascending sequence in the manuscript.
As per the literature review, all the figures under the results section must explain the findings and then compare them with previous models used for a similar study.
The paper contains potential, but it can be more beneficial and acceptable with the aforementioned changes.
Authors must complete the reviews and modify the findings per the above sections.

Cite this review as

Reviewer 2 ·

Basic reporting

1. The formatting is wired. Seems like a rejected version from another journal directly sent to PeerJ CS.

2. The writing needs further improvement. For example, in line 107, normally we don't include question mark in the research articles.

3. In line 470. What does 'Add your discussion here' mean??? This manuscript is absolutely not prepared.

Experimental design

The experimental design is a huge mess, did not deliver a clean introduction.

Validity of the findings

No comment

Cite this review as

Reviewer 3 ·

Basic reporting

The quality of the figures is very bad. Please pay special attention and improve the revised version.
The sequence of operations within the system could be better organized.
A brief explanation of the broader context could benefit the document.
The motivation, contribution, and benefits should be added in the introduction section.
The paper organization should be the part of introduction section.
The conclusion section should be supported by future work.

Experimental design

A brief explanation of the broader context could benefit the document.
What specific issues with network analysis and natural language processing does the proposed system aim to address?
The abstract should be modified by adding the implication, results of experimental results, and a brief overview proposed system.
The simulation environment, parameters, and tools should be mentioned in the results section.
What are the key functionalities of each of the three levels in the proposed system?
What makes this proposed system innovative compared to other existing solutions? What unique features or improvements does it offer?
Please add more results and compare them with state-of-the-art models.

Validity of the findings

What types of network analysis does the system perform, and what insights does it aim to provide?

How was the system tested or validated? Were any real-world scenarios used to evaluate its performance?

What makes this proposed system innovative compared to other existing solutions? What unique features or improvements does it offer?
How does the proposed system handle large-scale data? Are there any limitations in terms of scalability or performance?
What specific issues does the proposed system aim to address?

Cite this review as

---

## Round 0.2 · Minor Revisions

· Academic Editor

Minor Revisions

As per the reviewers' comments, many of their comments have not been incorporated into the manuscript. You are required to go through all the previous comments as well as new comments, revised your manuscript, and resubmit.

Reviewer 1 ·

Basic reporting

Similar issues persist in the updated manuscript as well.

The manuscript lacks proper English, so it is highly advised to have the proofreading completed by professional services or fluent English support speakers.

The Literature Review should introduce the State of Art Section in an organized way.

The figures submitted by the authors are not precise and per the Journal Guidelines. The figures should be modified for clarity of vision and understanding into vector graphics.

Several terms in the Formal results do not include the definitions used.
It is highly recommended that you update these terms and their meanings.

Experimental design

The submitted xlsm file seems to have some errors and is not working fine for retesting.
Authors must check it and update it.
A clear understanding of Corpus cleaning in the Title and abstract clean file is missing from the manuscript. The authors must explain the procedure clearly.

Validity of the findings

It is highly advisable to:
Several more potential keywords should be identified from the various analysis techniques the authors tried to include in this study.

Additional comments

As per the literature review, all the figures under the results section must explain the findings and then compare them with previous models used for a similar study.

The state-of-the-art comparison is still missing.
The authors have prepared a section introduction with new references, but the comparison with the latest ideas is still missing.

The authors must introduce the comparison to give strength to the paper.

Cite this review as

Reviewer 2 ·

Basic reporting

The writing and formatting, also the resolution for figures are still not improving.

Experimental design

The rational of the experimental design is still confusing.

Validity of the findings

This work is lack of novelty,

Cite this review as

Reviewer 3 ·

Basic reporting

The authors improved paper. However, curretnlt, intrdouction section is divided into different sections. it is not suggested. The suggestion is that add , Motiviation, contributtaion and the benfits of this research in the intrdouction; 1.1. motivation of this research, 1.2 our contributation etc.
In addation, The paper orangzation is still missing the intrdouction section. you should write as ; the rest of paper is organizaed as follows; we had discussed related work in section 2 etc.
Also check for the english and gramer mistakes etc.

Experimental design

It is fine

Validity of the findings

it is improved

Cite this review as

---

## Round 0.3 · accepted · Accept

· Academic Editor

Accept

I am pleased to inform you that your paper has been accepted for publication in PeerJ Computer Science. Your manuscript has undergone rigorous peer review, and I am delighted to say that it has been met with praise from our reviewers and editorial team. On behalf of the editorial board, I extend our warmest congratulations to you.

Reviewer 1 ·

Basic reporting

The authors have reported all the improvements as per the expectation.
The paper is currently acceptable in its initial format.

Experimental design

The authors have reported all the improvements as per the expectation.
The experimental design is eligible to be published.
The paper is currently acceptable in its initial format.

Validity of the findings

The authors have reported all the improvements as per the expectation.
The validity of findings is correctly reported.
The paper is currently acceptable in its initial format.

Additional comments

The authors have reported all the improvements as per the expectation.
The English Language is appropriately revised.
The authors have enhanced the quality of figures as per journal guidelines.
The paper is currently acceptable in its initial format.

Cite this review as

Reviewer 3 ·

Basic reporting

The authors improved the paper in this round. However, the image quality needs improvement, still few figures are not clear and readable.

Experimental design

It is improved.

Validity of the findings

It is fine.

Cite this review as